# Embryonic Development of Parthenogenetic and Sexual Eggs in Lower Termites

**DOI:** 10.3390/insects14070640

**Published:** 2023-07-15

**Authors:** Xin Peng, Zahid Khan, Xiao-Min Liu, Shi-Lin Deng, Yong-Gang Fang, Min Zhang, Xiao-Hong Su, Lian-Xi Xing, Xing-Rong Yan

**Affiliations:** 1College of Life Sciences, Northwest University, Xi’an 710069, China; 15209247629@163.com (X.P.); khanzahid370@yahoo.com (Z.K.); lxm990513@163.com (X.-M.L.); dengshilin00@163.com (S.-L.D.); yahoofangyonggang@163.com (Y.-G.F.); zhangmin19941222@163.com (M.Z.); sxhnwu@nwu.edu.cn (X.-H.S.); 2Zoology Department, University of Swabi, Swabi 23561, Khyber Pakhtunkhwa, Pakistan; 3Shaanxi Key Laboratory for Animal Conservation, Northwest University, Xi’an 710069, China; 4Xi’an Brand of Chinese Academy of Sciences, Xi’an 710043, China

**Keywords:** *Reticulitermes*, parthenogenetic eggs, single-cell transcriptome sequencing, embryo development, gene expression

## Abstract

**Simple Summary:**

It is important to compare the developmental stages of lower termites and recognize the differences between parthenogenetic and sexual eggs in order to describe their embryonic development concisely. In contrast with sexual eggs, parthenogenetic eggs have a lower rate of development and hatching success. Moreover, several genes differentially expressed between the two types of embryos have been identified. By addressing such a problem in the current research, the lack of understanding in the differences and the developmental processes between the two categories of eggs can be resolved. The aim of the study was to evaluate the development, discover the differences, and determine the gene expression patterns related to the variances between the two types of eggs. The authors discovered a number of genes with differential expression between the two categories of embryos, which explain these differences. These results present important insights into the reproductive processes of termites and highlight the need to contribute further research in this area to achieve a better understanding of termite biology.

**Abstract:**

Worldwide, termites are one of few social insects. In this research, the stages of embryonic development in the parthenogenetic and sexual eggs of *Reticulitermes aculabialis* and *R. flaviceps* were observed and described. In *R. flaviceps*, the egg development of the FF and FM groups happened during the early phases of development, whereas in *R. aculabialis*, this appeared mainly during the late phase of development. The variance in the number of micropyles between the *R. flaviceps* FF colony type and the *R. aculabialis* FF colony type was statistically significant. Five stages of egg development were found in both types of *R. aculabialis* but only the sexual eggs of *R. flaviceps.* In *R. flaviceps*, 86% of the parthenogenetic eggs stopped growing during the blastoderm development, with the yolk cell assembling frequently in the center of the egg. According to the results of the single-cell transcriptome sequencing, we investigated the egg-to-larval expression level of genes (*pka*, *map2k1*, *mapk1/3*, *hgk*, *mkp*, and *pax6*) and indicated that the levels of essential gene expression in RaFF were considerably higher than in RfFF (*p* < 0.05). We also discovered that the oocyte cleavage rate in the FF colony type was considerably lower in *R. flaviceps* compared to *R. aculabialis*, which gave rise to a smaller number of mature oocytes in *R. flaviceps*. During ovulation in both species, oocytes underwent activation and one or two cleavage events, but the development of unfertilized eggs ceased in *R. flaviceps*. It was shown that termite oocyte and embryonic development were heavily influenced by genes with significant expressions. Results from the databases KEGG, COG, and GO unigenes revealed the control of numerous biological processes. This study is the first to complete a database of parthenogenetic and sexual eggs of *R. flaviceps* and *R. aculabialis*.

## 1. Introduction

Lower termites can cause serious structural and agricultural damage to humans, as they only consume grass and wood, compared to higher termites that can consume soil and a range of other plant materials [1,2,3]. Under certain conditions, when males are unavailable, unfertilized eggs of a few termite species are able to develop into new female offspring via facultative parthenogenesis [3,4]. In China, *Reticulitermes aculabialis* and *R. flaviceps* are two economically significant species of subterranean termites. *R. aculabialis* can reproduce through facultative parthenogenesis [5], whereas *R. flaviceps* cannot. The development of individual oocyte cells can be separated into two distinct stages: (i) a preperiod of yolk arrangement; (ii) the yolk development stage [6,7]. In most insects, the development of oocytes stops after the primary meiotic metaphase Ⅰ [8], and is actuated again during ovulation when meiosis is completed [9] and the period of cleavage begins. During the normal pattern of superficial cleavage in insects such as *Drosophila*, the zygotic genome is interpreted and transcribed [10].

After the termite queen lay eggs, the workers transport the eggs to a concentrated egg pile for hatching. Some termite species have a short-germinal-type embryonic development [11]. Hence, studies on the embryonic development of *R. flavipes* have been conducted, dividing its embryonic development process into six stages according to morphological characteristics: cleavage and blastocyst formation, blastoderm and embryonic band formation, elongation and segmentation, the formation of a tail bend, rotation (embryo) and closure, and hatching [11]. While Matsuura et al. [12] studied the embryonic development of *R. speratus* and divided its embryonic development into five stages: fresh output, embryo formation, embryo extension, back healing, and complete development before hatching [12].

In mature oocytes, the *MPF* (development-promoting factor) regulates eukaryotic oocyte meiosis, the progress of both the G1/S and G2/M stages, and the MⅡ block [13,14]. The interaction between *MPF* and mitogen-activated protein kinase (MAPK), two protein kinases, has a very important effect on the meiotic maturation process of oocytes throughout the animal. MAPK during oocyte maturation, MII stagnation, and other aspects plays an important role, and it is activated during oocyte maturation [15]. In *Drosophila*, the *MAP2K1* gene could be involved in the meiosis of oocytes. The *PKA* gene is a cAMP-dependent protein kinase, and in the absence of cAMP, the *PKA* gene can exist in an inactive holoenzyme state. Previous studies have shown that *PKA* regulates mouse embryonic development and limbhog-induced limb development in *Drosophila*. The gene *CDK2* plays a major role in DNA replication, cell cycle regulation, and the transcriptional processes. It is a cell-cycle-dependent kinase (cyclin-dependent kinases). The *CPEB* gene is a highly conserved cytoplasmic polyadenylation-binding protein that mediates mRNA cytoplasmic polyadenylation and the translation of RNA. It not only mediates cell differentiation and cell senescence, but also promotes polyadenylation-induced translation. The *Pax6* gene can participate in the regulation of multiple signaling pathways in *Drosophila* and has a very important regulatory effect on *Drosophila* embryo development and adult tissue and organ differentiation [16,17].

Transcriptome sequencing technology can study all mRNAs transcribed from a specific tissue or cell at a certain time, detect gene expression in a genome-wide range, and perform differential gene screening analyses. It has the characteristics of high reproducibility, a wide detection range, and quantitative accuracy [18,19]. Single-cell transcriptome sequencing technology can amplify and sequence the entire transcriptome at the single-cell level. The principle is to amplify the microtranscriptome RNA of a single cell and then to perform high-throughput sequencing after amplification. This technology can reveal the gene expression status and gene structure information of the whole level in a single cell, accurately reflect the heterogeneity between cells, and allow for a deeper understand of the relationship between the genotype and phenotype. In recent years, with the development of high-throughput sequencing technology, transcriptome sequencing technology has been widely used in basic research, drug development, and clinical diagnosis [20,21,22,23,24], and the research method is more suitable for species whose genome maps are not yet completed or lack data information [25,26]. Haroon et al. [27] examined the age of different castes in *R. chinensis* by finding results through the databases KEGG, COG, and GO unigenes that concern the control of numerous biomechanisms.

In contrast to common perceptions, we propose a novel hypothesis in this article: that the eggs of asexual colonies via pairs of female–female alates are superior to the eggs of sexual colonies via pairs of female–male alates in *R. aculabialis* and *R. flaviceps*. Furthermore, for the egg developmental stages, molecular, morphological, and cytological evidence and advantages of the different development strategies are compared using these measures of parthenogenetic egg developmental success compared with sexual egg development in both species, such as blastoderm formation, the genetic control mechanism of egg development, fecundity, and the rate of development in two types of eggs of both species.

Facultative parthenogenesis demonstrates reproductive flexibility in some species of termites by allowing females to still reproduce with low mating rates or when males are absent. To date, little is known on oocyte development and the related genes in facultative parthenogenetic termites. The aim of our investigation is to measure the differences (morphological, histological, anatomical, and physiological) between the sexual reproduction and facultative parthenogenesis of the two *Reticulitermes* termite species in China, as well as to find out the reasons for the differentiation between the facultative parthenogenesis of *R. aculabialis* and non-parthenogenetic *R. flaviceps* in oocyte development. To understand the importance of sexual and parthenogenetic reproduction through the differential quantity of eggs, embryonic development, and gene regulation of different processes in the two termite species.

## 2. Materials and Methods

### 2.1. Establishment of Female–Female Colonies and Female–Male Colonies in the Laboratory

Wild colonies of *R. aculabialis* were collected from Xi’an and Nanjing during October 2015. Wild colonies of *R. flaviceps* were collected from Chengdu during April 2016. The male imagoes were separated from the female imagoes by using the differences in the shape of the seventh sternum as a diagnostic characteristic [12]. For comparison, four experimental colony types were established (200 bottles per colony type): (i) RfFF (*R. flaviceps* female–female counterpart group); (ii) RfFM (*R. flaviceps* female–male counterpart group); (iii) RaFF (*R. aculabialis* female–female counterpart group); (iv) RaFM groups (*R. aculabialis* female–male counterpart group). Each experimental colony was kept in a plastic container (π × 162 × 563 mm) with wet pine wood saw dust at 25 ± 5 °C in total darkness.

### 2.2. Ovary Photography

All termites were reared for 30 days. Thirty females were selected from each of the four colony types (RfFF, RfFM, RaFF, and RaFM) and transferred to slides with PBS solution. The whole female reproductive system was then removed and carefully dissected. Three-dimensional photographs were taken using a digital imaging microscope (Keyence VHX-5000, Keyence Corporation, Osaka, Japan) [28], observing the maturity levels of the ovaries and oocytes.

### 2.3. Micropyle Measurement

From each colony type (RfFF, RfFM, RaFF, and RaFM), 30 eggs were collected and used for the micropyle measurement using photographs and a digital imaging system (Keyence VHX-5000). The quantity of micropyles on each egg was counted and the diameter of each micropyle was measured. These statistical data were analyzed using SPSS 18.0 (SPSS Inc., Chicago, IL, USA).

### 2.4. Egg Nucleus Staining

During the regular observation of the spawning of the RfFF and RaFF colony types, eggs were taken every two hours, with the newest eggs reared by workers of the same species in a Petri dish with wet filter paper and cotton (egg: worker = 1:5); eggs at various developmental stages were removed for further investigation (30 eggs were taken for each developmental stage). The division of the egg developmental stages was observed at 0 h, 5 h, 10 h, 24 h, 48 h, 5 d, 10 d, 15 d, and 20 d.

The eggs collected from each stage were put into a 1.5 mL centrifuge tube, allowed to settle for 24 h within the modified Smith fixed liquid (0.5% K2Cr2O7:HCHO:CH3COOH = 35:4:4), flushed with PBS solution three times, and then soaked in 10% potassium hydroxide solution for one minute. Finally, the softened eggs were transferred into a PBS solution, where the eggshells were carefully removed manually using an anatomical needle with the aid of a stereomicroscope (Leica Zoom 2000, Shenzhen, China). Each stripped egg was put on a glass slide with a PBS solution and photographed. The stripped eggs were washed with PBS solution three times and stored in a methanol solution at 24 °C overnight. For dyeing, 0.2 μg/mL of DAPI liquid (methanol solution:DAPI stock solution = 4999:1) was then added and shaken at room temperature under darkness for 25 min. The stained eggs were shaken and rinsed three times for five minutes with a PBS solution. During the third rinse, a fluorescence quenching agent was added to the PBS solution. The eggs were examined under a cofocal laser microscope (Olympus FV1000, Shanghai, China) and the egg nuclei were counted by using the software Image-pro-plus 6.0 (Olympus, Shanghai, China).

### 2.5. Paraffin Section of Termite Eggs

The collected eggs from each stage were fixed with alcohol-based Bouin’s fluid (a saturated alcohol-based solution of picric acid, formalin: acetic acid = 15:5:1) at room temperature for 24 h. All the fixed eggs were stored in 70%, 80%, or 90% ethanol for 10 min, respectively. All eggs were then dried through exposure to desiccating agent A (anhydrous ethanol:ice acetic acid: chloroform = 6:1:3) and dehydrated with dehydrating agent B (positive butanol:ice acetic acid:chloroform = 6:1:3) for 20 min. Additionally, the dehydrated eggs were treated in sequence in terpinol I, II, and terpinol plus wax, wax I, and wax II. The fixed eggs were placed in a paraffin-filled mold and then removed after the paraffin had cooled and solidified. Paraffin sections were obtained after processing with a microtome (Leica RM2016, Leica Instruments Ltd., Shenzhen, China). The sections were stained with Delafield’s hematoxylin and eosin or with Azan [29], dried in a critical-point drier, and examined under a digital imaging system (Keyence VHX-5000, Keyence Corporation, Osaka, Japan).

### 2.6. Collection and Processing of Single-Cell Transcriptome Sequencing Materials

During the regular observation of the laying of the RfFF, RfFM, RaFF, and RaFM colony types, eggs were taken every two hours, with the newest eggs reared by workers of the same species in a Petri dish with wet filter paper and cotton (egg:worker = 1:5). After this, the removed eggs were cultured for 4 days in culture dishes; the eggs were immersed in an antibiotic solution for 4–6 h to sterilize the eggs [30] (note: antibiotic solution: 0.6 mg/mL penicillin, 1mg/mL streptomycin, 40 μL/mL gentamicin, and 10 μL/mL amphotericin B). After the completion of the sterilization, the eggs were lysed in a centrifuge tube containing 4 μL of a lysate (note: lysis: 10× lysis buffer:RNase inhibitor = 19:1). The eggs were gently punctured with an anatomical needle, which was sterilized and treated with de-RNase, so that the substances in the eggs were fully released.

### 2.7. Single-Cell Transcriptome Sequencing Process

Total RNA was extracted from each sample using Trizol Reagent (Invitrogen, Shanghai, China). Each sample’s total RNA was quantified using an Agilent 2100 Bioanalyzer (Agilent Technologies, Palo Alto, CA, USA), a NanoDrop (Thermo Fisher Scientific Inc., Tianjin, China), and an 11% agarose gel. The library preparation described as below utilized 1 g of an entire RNA that had an RIN value larger than 7. Following the manufacturer’s protocols for the NEBNext^®^ UltraTM RNA Library Prep Kit for Illumina^®^ (Genewiz, Tianjin, China), next-generation sequencing library preparations were created following the detailed and described protocol used by Haroon et al. [27] in their research study.

### 2.8. Data Analysis

To obtain high-quality clean data, pass-filter data in the fastq format were processed using Cutadapt [31] (version 1.9.1). Firstly, clean data were assembled using Trinity [32], which represents a novel method for the efficient and robust de novo reconstruction of transcriptomes from RNA-seq data. Trinity combines three independent software modules: Inchworm, Chrysalis, and Butterfly. These modules were applied sequentially to process large volumes of RNA-seq reads. Secondly, we removed the duplicated contigs using CD-hit, and then obtained the unigene sequence file.

With the unigene sequence file as the reference gene file, RSEM [33] (v1.2.6) estimated genes and isoform expression levels from the pair-end clean data. For the differential expression analysis, we used the DESeq [34] Bioconductor package, a model based on a negative binomial distribution. After adjusting with Benjamini and Hochberg’s approach for controlling the false discovery rate, the *p*-values of the genes were set to <0.05 to detect differentially expressed ones. GO-Term Finder was used to identify gene ontology (GO) terms that annotated a list of enriched genes with a significant *p*-value of less than 0.05. The KEGG (Kyoto Encyclopedia of Genes and Genomes) is a collection of databases dealing with genomes, biological pathways, diseases, drugs, and chemical substances (http://en.wikipedia.org/wiki/KEGG (accessed on 8 July 2023)). We used in-house scripts to enrich significant differential expression genes in the KEGG pathways and used the blast software to annotate the unigene sequence. All the databases included Nr, COG, Swissprot, KEGG, and GO. Through these three software programs, Misa, mreps, and trf, we obtained their individual SSR results and then kept where they intersected as the SSR final results.

### 2.9. RNA Isolation, cDNA Library Construction, and Quantitative Real-Time PCR

The total RNA of ten eggs in *R. aculabialis* and *R. flaviceps* female–female colony types was extracted using TRNzol-A+ Reagent (Tiangen, Beijing, China), according to the manufacturer’s protocol. RNA quality was verified using a spectrophotometer (Eppendorf BioPhotometer, Shanghai, China). cDNA for qPCR was synthesized using Primer Script RTase (TaKara Bio. Inc., Hong Kong, China). The quantitative reaction was performed with LightCycler 480 software release 1.2.0.0625 (Roche Diagnostics, Indianapolis, IN, USA) using SYBR Premix Ex TaqTMⅡ (TaKara Bio. Inc., Baori Medical Biotechnology Beijing, China). All reactions in the qPCR system were normalized using the Ct values corresponding to the actin levels according to a study of reliable reference genes for expression studies using qPCR in *Odontotermes formosanus* [35]. The relative gene expressions were calculated using the 2−∆∆Ct method [36] and the data were analyzed with SPSS 18.0. All qPCR experiments were repeated for three technical replications. The fragment size of the primer was between 100 and 250 bp, with details presented in Table 1.

## 3. Results and Analysis

### 3.1. Ovary Measurements

The dissected ovaries, oocytes, and their maturity levels were examined and compared with the previously published literature [37]. As a result, mature ovaries were divided into five major developmental levels (A–E) (Figure 1). A: The preyolk formation stage of oocytes; a short ovarian tube size and denser clusters. Only preyolk formation oocytes were observed (oocyte length < 0.18 mm) and no yolk formation oocytes were found (Figure 1(A1,A2)). B: Early yolk formation; an ovarian tube with one or more early yolk formation oocytes (0.18–0.32 mm); no late and mature yolk formation oocytes were found (Figure 1(B1,B2)). C: Late yolk formation; there were one or more late yolk formation oocytes (0.32–0.58 mm) in the ovarian tubes, which were mostly branched, but no mature oocytes were present (Figure 1(C1,C2)). D: Mature oocyte stage; there were one or more mature basal oocytes in the ovary tubes (0.58–0.80 mm) (Figure 1(D1,D2)). E: Oocyte stage of the lateral oviduct; mature oocyte moved from the ovarian tube and entered the lateral oviduct (Figure 1(E1,E2)).

Based on these ovarian developmental stages, the FF and FM groups of *R. flaviceps* and *R. aculabialis* were compared, obtaining the proportion of the number of samples for each stage (Figure 2). The proportions of the FF group of *R. flaviceps* were A (13.33%), B (40%), C (33.33%), D (6.67%), and E (6.67%) (Figure 2). Those of the FM group of *R. flaviceps* were A (25%), B (33.33%), C (25%), and D (16.67%). Moreover, the FF group for *R. aculabialis* proportions were C (41.67%) and D (58.33%) stages, whereas no samples contained ovaries of stages A and B. The proportions of the FM group for *R. aculabialis* in each stage were: A (8.33%), B (8.33%), C (33.33%), D (41.67%), and E (8.33%). Furthermore, the ovarian development of the FF and FM colony types of *R. flaviceps* was in the early developmental stages (the oocyte stage of preyolk formation or the oocyte stage of early yolk formation and stages B and C). The ovarian development of the FF and FM types of *R. aculabialis* was mostly in the late developmental stages (midyolk formation stages of oocytes and mature oocytes, stages C and D).

### 3.2. Micropyles

Micropyles are on the posterior of termite eggs and are arranged in an arc at unequal intervals (Figure 3A); individual micropyles are funnel-shaped (Figure 3B). The micropyle numbers per egg in the FF and FM colony types in *R. flaviceps* were 6.31 ± 1.89 and 8.18 ± 3.22, respectively, whereas in *R. aculabialis*, the micropyle numbers of FF and FM egg types were 8.15 ± 2.66 and 8.43 ± 3.05, respectively. A single-factor analysis of variance showed that the difference was significant in micropyle numbers between the *R. flaviceps* FF colony type and the *R. aculabialis* FF colony type (*p* < 0.05). There were no significant differences in the numbers of micropyles between other groups (*p* > 0.05) (Figure 4).

### 3.3. The Development of Parthenogenetic Embryos in R. aculabialis

The development of parthenogenetic embryos was observed in *R. aculabialis*, and the embryonic development concurred with the typical short-embryo model investigated in the parthenogenetic eggs of *R. aculabialis*. During development, the embryo was placed on the surface of the yolk and was not trapped in the yolk. Embryonic development was divided into five stages based on morphological observations; those are shown in Table 2.

First stage: oocytes in the ovaries. The length of the oocytes was approximately 605.8 μm, and the accumulation of yolk was nearly complete, forming a single layer of follicle cells outside each oocyte. Due to the presence of follicular cells, the egg nucleus could not be directly observed (Figure 5a(B)). Yashiro and Matsuura [38] divided the egg formation into seven phases, while a similar number of embryonic stages were also reported in the present research work of the above-mentioned termites. Figure 5a(C) shows the seventh phase before ovulation; the length of the oocytes was approximately 685 μm, and the follicular cells were disappearing and forming the eggshell. Only one nucleus was seen in the mature oocytes before ovulation after the eggshell was peeled (Figure 5a(C)).

Second stage: cleavage and blastoderm formation. Here, termites were shown with a superficial cleavage of oocytes [39]. Directly after oviposition, the number of egg nuclei was 2–5 (Figure 5a(E)), which suggested that the eggs of *R. aculabialis* in the FF colony type had cleaved inside the termite prior to oviposition, and that the egg activation occurred during ovulation. The egg continued to cleave after oviposition, and the resulting cells gradually moved towards the surface of the egg. Five hours after the oviposition, the eggs contained 16–22 nuclei (Figure 5a(F)). At 10 and 24 h postoviposition, the nuclei had increased to approximately 35–42 (Figure 5a(G)) and 51–60 (Figure 5a(H)), respectively. At 48 h later, approximately 67–84 nuclei were present, most of which moved to the back of the egg (Figure 5a(I)). The cleavage rate increased within the first five hours, followed by a gradual reduction. After 3–5 days, the blastoderm became visible at the back end of the egg (Figure 5a(J,K)).

Third stage: germinal band formation, extension, and segmentation. On the surface of the posterior yolk mass, a curved embryonic band was visible (Figure 5a(L,M)); the anterior end of the early germ band remained motionless, and the posterior end extended around the back convex surface of the egg, forming a “U” shape around the posterior end (Figure 5a(N,O)). The head then expanded to both sides and formed the forehead lobe (Figure 5b(P,Q)). At the same time, the embryo continued to extend backwards. When the embryo extended to half of the egg’s length, the head appendages and the middle segmentation of the embryo became visible (Figure 5b(R,S)).

Fourth stage: bend in the tail formation and embryo motion. With the elongation of the embryo, the end of the embryo formed a double-curved structure, and the appendages at the head and thorax continued to elongate, with the abdominal segments becoming visible (Figure 5b(T)). The embryo rotated twice. The first rotation was around the short axis of the egg; the head part moved to the front end along the protruding side of the egg, while the tail part moved to the posterior end along the sunken side of the egg. The double-curved structure disappeared, with only a single curve remaining (Figure 5b(U)). During the second rotation, the embryo rotated 180° around the longitudinal axis of the egg. The dorsal organ of the embryo gradually sank into the yolk and the embryo rotated around the longitudinal axis, reversing the position of the back and abdomen so that the embryo’s foot faced the concave surface of the egg (Figure 5b(V)).

Fifth stage: back closure and embryonic organ development. After two rounds of rotation, the germ band extended bilaterally and dorsally along the periphery of the remaining yolk mass until the edge of the germinal band merged at the center of the dorsum and formed a cavity (Figure 5b(V,W)). At the same time, the size of the embryo increased rapidly, and internal organs gradually formed (Figure 5b(X,Y)).

### 3.4. The Development of Parthenogenetic Embryos in R. flaviceps

The mature oocytes in the ovaries of *R. flaviceps* shown in Figure 6A were approximately 657 μm in length. Figure 6B shows the DAPI-stained mature oocytes in the ovaries of *R. flaviceps*, from which the eggshells were removed. Only one egg nucleus was visible in each oocyte, as shown Figure 6C. This also showed the external forming process of the cleavage and that the embryonic volume did not increase during early cleavage, so any external morphological differences were not obvious. The number of egg nuclei was 6–10 at 5 h postoviposition and 28–37 at 48 h postoviposition. At 5–7 days postoviposition, the number of egg nuclei ranged from 71 to 85. During the cleavage process, the nuclei moved to the posterior end of the egg, where they accumulated. After approximately 10 days, approximately 86% of the embryos stopped developing at the blastoderm formation period, and approximately 10% of the embryos stopped developing at the embryonic formation period. The yolk cells were large, gathering in the center of the eggs, with a hollow cavity on both sides of the yolk membrane. Additionally, 4% of the embryos occupied the entire eggs, but lacked any differentiation in appendages and organs. All the mentioned observations can be seen in Table 2.

### 3.5. Single-Cell Transcriptome Sequencing of Parthenogenetic and Sexual Eggs of R. aculabialis

The library assembly of parthenogenetic and sexual eggs of *R. aculabialis* was 324,454, the total number of bases was 170,772,245, the shortest unigene length was 201 bp, and the longest unigene length was 13,549 bp. The average length of all unigenes was 526.34 bp (Table 3). There were 225,450 transcripts with a length of 200–500 nt and a ratio of 69.49%; there were 67,067 transcripts with a length of 500–1000 nt and a ratio of 5.13%; there were 16,649 transcripts with a length of 1000–1500 nt and a ratio of 5.13%; there were 6909 transcripts with a length of 1500~2000 nt and a ratio of 2.13%; there were 8381 transcripts with a length of more than 2000 nt, accounting for 2.58% (Figure 7). The Nr database was annotated to 61,289 unigenes, the Swissprot database was annotated to 24,808 unigenes, the KEGG database was annotated to 2,354 unigenes, the COG database was annotated to 23,507 unigenes, and the total number of unigenes annotated by the four databases was 62,253 (Figure 8).

Unigenes’ GO annotation results covered three broad categories; molecular function, cellular components, and biological processes. The top ten unigenes involved in the biological processes were catalytic activity, binding, the metabolic process, cellular process, cell part, single-organism process, membrane part, biological regulation, organelle, and macromolecular complex (Figure 9). The unigenes annotated with the KEGG database included five major categories (primary classification) of cellular processes, environmental information processing, genetic information processing, human diseases, and metabolism. There was a breakdown of specific functional categories (secondary categories) under each broad category. Among them, the transcripts involved in signal transduction were the most prevalent, belonging to the category of environmental information processing; the second were the transcripts involved in the transition, which belonged to the category of genetic information processing. The transcripts participating in the endocrine system ranked third, belonging to the biological system category (Figure 10). In the COG database function classification, the comment had the most function in the signal transduction mechanism function, followed by the general function prediction only (Figure 11).

A total of 11,892 simple repeats (SSRs) were identified in unigenes, distributed in 11883 sequences. Among them, dinucleotides were the most important form of microsatellite expression, of which there were 1236, accounting for 10.39%; the number of trinucleotides was 735, accounting for 6.18%; the number of tetranucleotides was 258, accounting for approximately 2.17%; and the number of pentanucleotides was 13, accounting for approximately 0.11% (Table 4).

The results of the test were screened according to the difference significance criterion (more than twice the differential genes changed multiple times and a *p* value ≤ 0.05). The number of differential genes between parthenogenetic and sexual eggs of *R. aculabialis* was 987, of which 710 were upregulated and 277 were downregulated (Figure 12). The GO enrichment analysis of differential genes revealed that they were mainly concentrated in the three functions of catalytic activity, binding, and metabolic processes (Figure 13). The KEGG enrichment analysis of differential genes revealed that it was mainly enriched in the proteasome pathway and was not significantly enriched in other pathways (Figure 14). The COG analysis of differential genes revealed that the differential genes were mainly concentrated in the three functions of general function prediction only, post-translational modification, protein turnover, chaperones, and function unknown (Figure 15).

### 3.6. Single-Cell Transcriptome Sequencing of Parthenogenetic and Sexual Eggs of R. flaviceps

The number of unigenes assembled in the library of termite parthenogenetic and sexual *R. flaviceps* eggs was 422,213, the total number of bases was 202,603,877, the shortest unigene length was 201 bp, the longest unigene length was 42,295 bp, and the average length of all unigenes was 479.86 bp (Table 5). There were 315,080 transcripts with a length of 200–500 nt and a ratio of 74.63%; there were 77,347 transcripts with a length of 500–1000 nt and a ratio of 18.32%; there were 15,598 transcripts with a length of 1000–1500 nt and a ratio of 3.77%. There were 6349 transcripts with a length of 1500–2000 nt and a ratio of 1.5%; there were 7538 transcripts over 2000 nt, accounting for 1.79% (Figure 16). The Nr database was annotated with unigenes for 122,100 unigenes, the Swissprot database was annotated to 48,416 unigenes, the KEGG database was annotated to 11,086 unigenes, the COG database was annotated to 36,805 unigenes, and the four databases were annotated with unigenes for 5781 unigenes (Figure 17).

In unigenes’ GO annotation results, the top ten numbers of unigenes involved in biological processes were catalytic activity, binding, the metabolic process, cellular process, cell part, single-organism process, membrane part, biological regulation, membrane, and organelle (Figure 18). The KEGG database had the largest number of transcripts involved in global and overview maps in unigenes, followed by transcripts involved in the carbohydrate metabolism, with transcripts involved in the amino acid metabolism ranked third, belonging to a large class of metabolisms (Figure 19). In the COG database functional classification, annotations were the most common in the general function prediction only function, followed by signal transduction mechanisms, transcription, post-translational modification, protein turnover, chaperones, translation, ribosomal structure and biogenesis, amino acid transport and metabolism, lipid, transport and metabolism, energy production and conversion, function unknown, etc. (Figure 20).

A total of 7196 simple repeats were identified in indigene distributed over 7186 sequences. There were 588 dinucleotides, accounting for 8.17%; the number of trinucleotides was 420, accounting for 5.84%; the number of tetranucleotides was 143, accounting for approximately 1.99%; the number of pentanucleotides was 10, accounting for approximately 0.14% (Table 6).

The test results were screened according to the difference significance criterion (more than two times the change in differential gene expression and a *p* value ≤ 0.05) and the number of differential genes between parthenogenetic and sexual eggs of *R. flaviceps,* which was 19,153. Of these, 484 were upregulated and 18,669 were downregulated (Figure 21). The GO enrichment analysis of differential genes revealed that they were mainly distributed in three functions, namely, catalytic activity, binding, and metabolic processes (Figure 22). The KEGG enrichment analysis of differential genes revealed no results in the analysis. The COG analysis of differential genes revealed that the differential genes were mainly concentrated in three functions, namely, amino acid transport and metabolism, energy production and conversion, and general function prediction only (Figure 23).

### 3.7. QRT-PCR Verification of the Gene Expression Analysis of Embryonic Development

Eight differential genes were selected that associated with embryonic development for the real-time PCR sequencing (Figure 24). The analysis of the results revealed that there was no significant difference in the expression levels of the CDK2 gene between the groups. The expression levels of the PKA gene and the MAP2K1 gene in RaFF, RfFM, and RaFM were significantly higher than those in RfFF (*p* < 0.05). The level of expression of the PKA gene in RaFF was significantly higher than in RaFM and RfFM (*p* < 0.05). The expression levels of the MAP2K1 gene in RaFM and RfFM were significantly higher than those in RaFF (*p* < 0.05). The expression level of the CPEB gene in RaFF was significantly higher than that in RfFM (*p* < 0.05), and there was no significant difference between the other groups. The expression levels of the MAPK1/3 gene and HGK gene in RaFF and RfFM were significantly higher than those in RaFM and RfFF (*p* < 0.05), and there was no significant difference between the other groups. The expression level of the MKP gene in RaFF was significantly higher than that in RaFM, RfFM, and RfFF (*p* < 0.05), and the others had no significant differences. The expression level of the Pax6 gene in RaFF was significantly higher than that in RaFM, RfFM, and RfFF. The expression level in RaFM was significantly higher than that in RfFM and RfFF (*p* < 0.05), and the others had no significant differences.

## 4. Discussion

The ovarian development of the FF and FM colony types of *R. flaviceps* was in the early developmental stages (the oocyte stage of preyolk formation or the oocyte stage of early yolk formation), whereas the ovarian development of the FF and FM types of *R. aculabialis* was mostly in the late developmental stages (midyolk formation stages of oocytes and mature oocytes) (Figure 2). The ovaries of *R. aculabialis* virgin adult females demonstrated high levels of maturity. Similar findings were discovered in a higher level of investment in reproduction, visible in the storage of abundant proteins and various nutrients in *Drosophila* oocytes during embryogenesis [10].

Additionally, the *Drosophila* embryos possessed maternally derived cell regulators that could trigger the cleavage of early embryonic cells. As the maternal protein was insufficient for further embryonic development, mRNA stored during oogenesis was used to translate the cyclic regulators during the synthesis of the cleavage process, and the early embryonic development of eggs was inseparable from that of the accumulation of oocytes in the ovary [10]. Thus, the reduced cleavage rate of the parthenogenetic eggs in *R. flaviceps* may have been the result of insufficient mature oocyte cells when the ovary was still immature.

Hence, a number of micropyles were found on eggs laid by female–female colonies of both species in this study, which concurred with previous reports, in which micropyles were present on the parthenogenetic eggs of the termite *R*. *speratus* during the early colony stage [40,41]. Contrary to previous research, no micropyles were found on the unfertilized eggs of *R. chinensis* [41]. However, we found micropyles on the fertilized and unfertilized eggs of both *R. flaviceps* and *R. aculabialis* (Figure 4). The micropyle numbers of *R. aculabialis* counted in RaFF eggs and RaFM eggs were 6.31 ± 1.89 and 8.18 ± 3.22 3.22, respectively, while higher measurements were obtained for *R. flaviceps* RfFF and RfFM eggs, amounting to 8.15 ± 2.67 and 8.43 ± 3.05, respectively. The number of micropyle on RfFF eggs was higher than on RaFF eggs, but there was no significant difference between the other groups. This difference between *R. chinensis* and both other congeneric species may have been due to phylogenetic effects.

Asexual colonies were set up in pairs of female–female alates and sexual colonies were set up in pairs of male–female alates, examining the fecundity of parthenogenetic and sexual colonies, their fitness and success, their rate of growth and development, offspring, and the survival of *R. speratus* [2]. Here, similar results were observed in the parthenogenetic and sexual colonies of *R. aculabialis.* While parthenogenetic colonies of *R. flaviceps* showed numbers of eggs that had growth and development up to cleavage, they failed to form a regular blastoderm, germinal band, organs, and appendages.

Hence, fecundity related to *R. aculabialis* RaFF eggs and *R. flaviceps* RfFF eggs was, respectively, 35.12 ± 2.59 and 26.64 ± 3.78 after nest establishment, and the monthly mean number of collected eggs per single colony in August was 18.24 ± 3.18 and 11.53 ± 4.51, respectively, while in September it was 11.164.26 and 3.67 ± 1.24, respectively. The results showed that *R. flaviceps* parthenogenetically deposited a smaller number of eggs than *R. aculabialis* in a couple of months, while the *R. flaviceps* deposited lesser eggs in September than August. Our findings were consistent with those of numerous other researchers, such as Ishitani and Maekawa [42,43], who discovered that the number of eggs in early-stage female–female nests was substantially greater than that in female–male nests.

The egg size of *R. aculabialis* changed from 605.8 μm to 685 μm due to developmental mechanisms, and the egg size of *R. flaviceps* also grew from 657μm onward. It also changed with the developmental performance of the eggs as the embryo size increased due to the rapid formation of internal organs and appendages. However, it was discovered that parthenogenetic eggs were larger in size than sexual eggs, had longer incubation periods than sexual eggs, and the hatching rates of both types of eggs were similar in termite *R. speratus* [9].

Furthermore, by analyzing the development of the parthenogenetic embryos in *R. flaviceps* and *R. aculabialis* (Figure 5 and Figure 6), it was found that the cleavage rate of *R. aculabialis* slowed gradually around the fourth day, as investigated in *Drosophila* [10]. While the cleavage rate of newly produced eggs might have been higher due to the mitosis of the maternal mRNA protein, with the degradation of maternal mRNA, the rate of cleavage slowed and the egg initiated the nucleus genome to encode for new protein synthesis [10,41]. The egg development of *R. flaviceps* ceased in the blastoderm formation after approximately 8–10 days postoviposition. We suggest that after the degradation of the maternal mRNA, the genome of *R. flaviceps* was unable to begin encoding for new proteins, which would stop any further development of the embryo.

In the context of the yolk cells clustering in the middle of the egg and both ends of the egg being emptied during the late cleavage stage or at the blastoderm stage in *R. flaviceps*, we propose a conceivable assumption that this yolk cell clustering prevented expansion, while it was determined in moth *Bombyx mori* eggs during diapause that the yolk cells also amassed into clusters and began to spread within the egg after diapause ended. It was suggested that the yolk was a basic supplementary hotspot for embryonic development and likely played a critical role in the supply of egg nutrients [44]. In our observation, the yolk cells of *R. flaviceps* parthenogenetic eggs did not spread as the embryo developed, which may reflect a decrease in nutrients within an egg. This may explain, at least in part, the suspension of embryonic development in *R. flaviceps* parthenogenetic eggs.

Additionally, progeny in FF colonies was lower than in the sexual colonies of both termite species. Our results corresponded with the results of Matsuura et al. (2004), as the observed number of first-batch progeny in female–female nests was significantly lower than the number of offspring in female–male nests. The embryonic development of the parthenogenetic eggs of *R. flaviceps* and *R. aculabialis* (Figure 5 and Figure 6) provided evidential insights that the cleavage rate of *R. aculabialis* slowed progressively on approximately the fourth day as compared to sexual egg development. Both developmental and genetic conditions restricted parthenogenesis, and its progeny typically had a lower survival rate than those of sexual reproduction. Similar results were provided by Matsuura et al. (2004), who determined that parthenogens had a higher mortality rate and a longer period of egg development than sexually produced offspring.

A number of cytological analyses were carried out through the entire developmental processes to confirm the movement of the cells, nuclei, blastoderm, blastocoel, germinal band formation, organs, and appendages, although *R. flaviceps* and *R. aculabialis* development was classified into five phases, like in Matsuura et al.’s work [12]. Contrary to this study, Matsuura et al. [45] also performed a cytological analysis of male and female alates and their progeny, which showed that all of them had 2n = 42 somatic chromosomes.

The embryonic stages of termite development were crucial; during this time, the embryo was served by other individuals of the colony for successful development and hatching; even larvae W1 and W2 were fed by them. These findings were generally in agreement with the results reported in previous studies [2,36,42].

A single-cell sequencing analysis of the transcriptome consisting of parthenogenetic and sexual eggs of *R. aculabialis* and the transcriptome consisting of parthenogenetic and sexual eggs of *R. flaviceps* revealed that a large number of unigenes were not annotated in both transcriptomes. Most of the sequences did not match the protein sequences of other species, probably because the termites did not have a reference genome. The different gene between parthenogenetic and sexual eggs of *R. flaviceps* was significantly greater than that between parthenogenetic and sexual eggs of *R. aculabialis*, which could develop normally with *R. aculabialis* orphaned eggs and *R. flaviceps*. Morphological findings of developmental blockages were consistent. Eight differentially expressed genes related to embryonic development were selected from two single-cell transcriptome sequencing results for the real-time PCR analysis. The results of this sequencing were reliable.

The high expression of the *MAPK1/3* gene and *HGK* gene in RaFF contributed to the breakthrough of developmental arrest in *R. aculabialis* parthenogenetic eggs, and the process of cell proliferation continued to ensure the normal development of embryos. Therefore, it was hypothesized that the low expression of the *MAP2K1* gene and the *PKA* gene in RfFF inhibited cell proliferation, which caused *R. flaviceps* to parthenogenetically develop slowly on the third to fifth days, at which point the development finally stopped. Similarly, it was indicated in *Drosophila* that the *MAP2K1* gene is involved in oocyte meiosis, while *PKA* inhibited cell proliferation by blocking the activation of Ras or MAPK signaling pathways. *MAPK1/3* is a multichannel functional protein in the mitogen-activated protein kinase (MAPK) family [39]. The *HGK* genes in RaFF and RfFM were particularly higher than in RaFM and RfFM, whereas there was no difference between the other groups. These outcomes were consistent with previous discoveries that the *HGK* gene is involved in cell movement, cytoskeletal rearrangement, and cell proliferation, while *MKP* participates in the apoptotic process by regulating the MAPK signaling pathway [46].

In our results, there was a high expression of the *Pax6* gene and the *MKP* gene in RaFF, facilitating the development of lone female eggs in *R. aculabialis* and allowing parthenogens to develop normally. Similar results were determined for *Pax6*, having a very important regulatory effect on the development of *Drosophila* embryos and the differentiation of adult tissues and organs [46].

In the *R. flaviceps* parthenogenetic egg, there was no difference in the expression level of the *R. flaviceps* bisexual egg, which caused the *R. flaviceps* parthenogenetic egg to be unable to break through the developmental blockage. However, the gene expression of *cdk2*, *pka*, *map2k1*, *mapk1/3*, *hgk*, and *mkp* decreased progressively after the fourth day, which became the reason for the delayed development of the parthenogenetic eggs.

Longevity genes, such as *Pdk1*, *akt2-a*, *Tsc2*, *mTOR*, *EIF4E*, and *RPS6*, were investigated in different polymorphic categories of *R. chinensis* by obtaining results through the databases KEGG, COG, and GO unigenes, involved in the control of many biological mechanisms [27]. Contrarily, microsatellites were used to compare the genotypes of the parents and offspring in order to examine parthenogenesis. The reported mtDNA was the same among queen FF and progeny [45].

The expression of genes in *R. flaviceps* and *R. aculabialis* parthenogenetic embryos confirmed different phenotypes, such as cleavage, the distribution of yolk cells, and number of micropyles. This study provided gene database resources for further research on differential genes between the parthenogenetic and sexual eggs of *R. flaviceps* and *R. aculabialis*, as well as a reliable molecular research basis for revealing the mechanism of parthenogenesis in *R. aculabialis* and development retardation in *R. flaviceps*.

## 5. Conclusions

A morphological study of the parthenogenetic eggs of *R. flaviceps* and *R. aculabialis* in the early stages in a laboratory revealed that the parthenogenetic eggs of *R. flaviceps* could still develop embryos after ovulation, while the embryonic development of *R. flaviceps* parthenogenetic eggs showed a phenomenon of developmental arrest later. In this study, a single-cell transcriptome sequencing analysis of embryo development in parthenogenetic and sexual eggs of *R. flaviceps* and *R. aculabialis* was performed. The number of unigenes obtained by assembling the transcriptome composed of sexual eggs and parthenogenetic eggs of *R. flaviceps* was 422,213, the total base number was 202,603,877, the shortest unigene length was 201 bp, the longest unigene length was 42,295 bp, and the average length of all unigenes was 479.86 bp. The transcriptome of *R. flaviceps* was annotated with 123,667 unigenes, of which 122,100 were annotated in the database, the COG database with 36805 unigenes, the Swissprot database with 48,416 unigenes, and the KEGG database with 2354 unigenes. There were 7196 simple repeat sequences (SSRs) identified in the transcriptomes of sexual eggs and parthenogenetic eggs of *R. flaviceps* distributed in 7186 sequences: 588 dinucleotides, 420 trinucleotides, 143 tetranucleotides, and 10 pentanucleotides. The expression levels of *pka*, *map2k1*, *mapk1/3*, *hgk*, *mkp*, and *pax6* genes had a promoting effect on the normal development of parthenogenetic eggs of *R. aculabialis*. The development of parthenogenetic eggs in *R. flaviceps* was slow due to a low gene expression. In the five stages of embryonic development of RfFMeg, RaFFeg, and RaFMeg, the gene expression of *cdk2*, *pka*, *map2k1*, *mapk1/3*, *hgk*, and *mkp* reached its maximum level of expression on the second to fourth days. However, the gene expression level began to decline gradually after the fourth day. This suggested that the delayed development of parthenogenetic eggs of *R. flaviceps* starting from the third to the fifth days may have been due to the gradual decrease in gene expression levels of *cdk2*, *pka*, *map2k1*, *mapk1/3*, *hgk*, and *mkp* after the parthenogenetic eggs of *R. flaviceps* developed to the second day.

## Figures and Tables

**Figure 1 insects-14-00640-f001:**
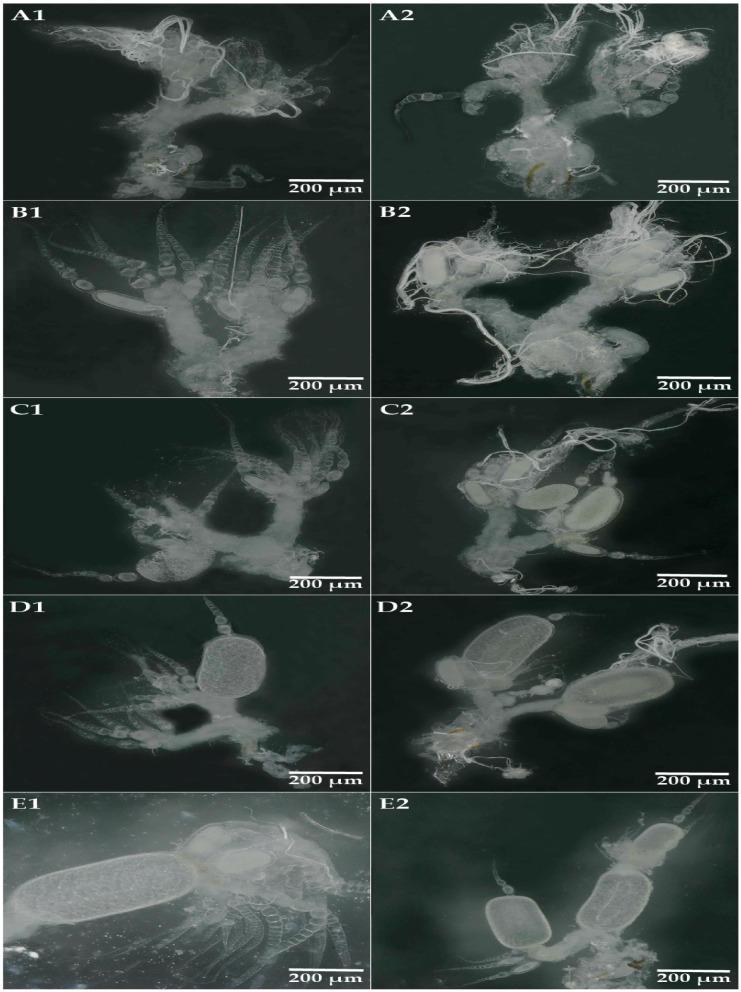
Different ovary development stages of *R. aculabialis* and *R. flaviceps* (**A1**,**B1**,**C1**,**D1**,**E1**): anatomical pictures of the female reproductive system of *R. flaviceps*. (**A2**,**B2**,**C2**,**D2**,**E2**): anatomical pictures of the female reproductive system of *R. aculabialis*. O: ovary. GC: genital chamber. OV: oviduct. Ag: female accessory gland. S: spermatheca. (**A**–**E**) represent the five developmental states of the ovaries. (**A**): preyolk formation stage of oocytes. (**B**): early yolk formation. (**C**): late yolk formation. (**D**): mature oocyte stage. (**E**): oocyte stage of the lateral oviduct.

**Figure 2 insects-14-00640-f002:**
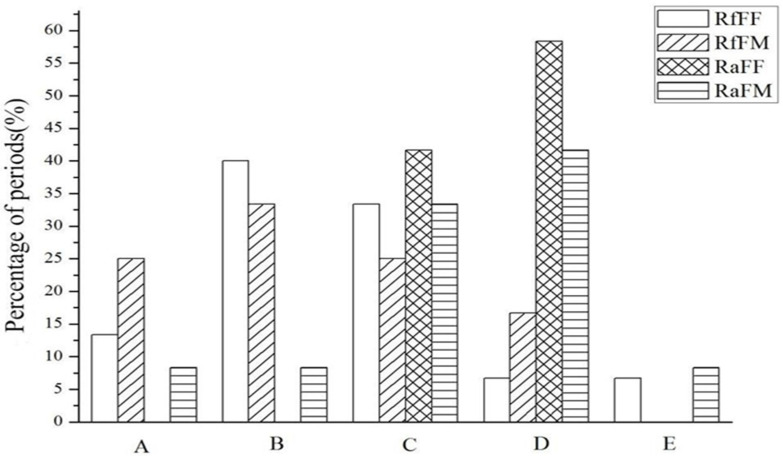
The percentage of ovary development at each stage in *R. aculabialis* and *R. flaviceps*. The five ovary stages. (A): preyolk formation stage of oocytes. (B): early yolk formation. (C): late yolk formation. (D): mature oocyte stage. (E): oocyte stage of the lateral oviduct. The graph ordinate is the proportion of the experimental group ovaries at some stage. RfFF: female adult genital organs in the FF group of *R. flaviceps*. RfFM: female adult genital organs in the FM group of *R. flaviceps*. RaFF: female adult genital organs in the FF group of *R. aculabialis*. RaFM: female adult genital organs in the FM group of *R. aculabialis*.

**Figure 3 insects-14-00640-f003:**
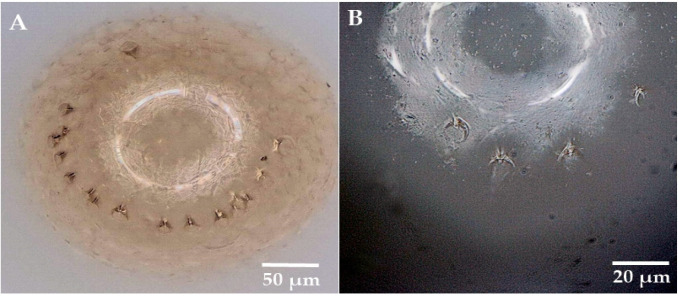
Morphology of micropyles in eggs of *R. aculabialis* (**A**): the convex view of the back end of the termite egg with the 17 micropyles arranged in an arc shape. (**B**): enlarged view of micropyles, showing the shape of the funnel.

**Figure 4 insects-14-00640-f004:**
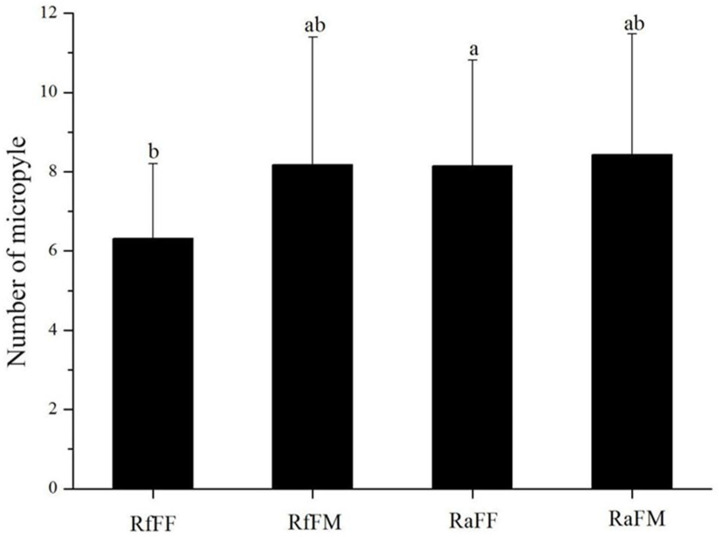
Comparison figure of micropyle numbers for *R. aculabialis* and *R. flaviceps*. Data are mean ± standard deviation. The different lowercase letters above the columns of the figure indicate significant differences among the experimental groups. The same letter indicates no significant difference (*p* > 0.05), and the different letters indicate a significant difference (*p* < 0.05). RfFF eggs of *R. flaviceps* FF group; RfFM eggs of *R. flaviceps* FM group; RaFF eggs of *R. aculabialis* FF group; RaFM eggs of *R. aculabialis* FM group.

**Figure 5 insects-14-00640-f005:**
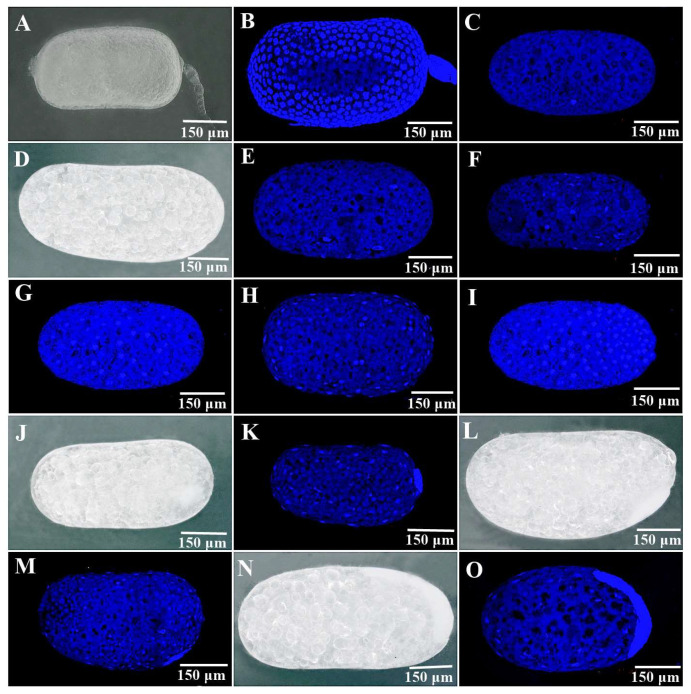
(**a**) The embryo development process of female–female eggs of *R. aculabialis* (**A**,**D**,**J**,**L**,**N**): the morphological forms of embryo development process with no staining at various stages of the egg laid by *R. aculabialis* FF group. (**B**,**C**,**E**–**I**,**K**,**M**,**O**): the morphological forms of embryo development process stained with DAPI at various stages of the egg laid by *R. aculabialis* FF group. (**A**–**C**): oocyte stage in the ovary. (**B**): a follicular cell outside the immature oocyte; the dark blue part shows the nuclei of the follicular cell. (**C**): mature oocytes peeled off the eggshell; the dark blue part shows the nuclei of oocytes. (**D**–**I**): oocyte cleavage phase stage. (**E**): 0 h after egg laying, there were 3 egg nuclei. (**F**): 5 h after egg laying, there were 18 egg nuclei. (**G**): 10 h after egg laying, there were 39 egg nuclei. (**H**): 24 h after egg laying, there were 58 egg nuclei. **I**: 48 h after laying, there were 79 egg nuclei. (**J**,**K**): 3–5 days after egg laying, the blastoderm started forming. (**L**,**M**): 7–8 days after egg laying, the germinal band started forming. (**N**,**O**): 10–15 days after egg laying, the germinal band started extending, forming a “U” structure located at the back end of the egg. (**b**) (**P**,**R**): the external form of embryo development (no staining) at various stages of the egg from FF group. (**Q**,**S**,**W**,**Y**): the external form stained with DAPI at various stages of the egg from FF group. (**T**–**V**,**X**): the paraffin slices of each period of the egg from FF group. P and Q: the head lobe widened. (**R**,**S**): appendages in the head and thorax and the abdomen segmentation. (**T**): the double-curved structure of the tail part. (**U**): first rotation around the short axis. (**V**,**W**): second rotation around the long axis, completing the back healing. (**X**,**W**): development of appendages and internal organs.

**Figure 6 insects-14-00640-f006:**
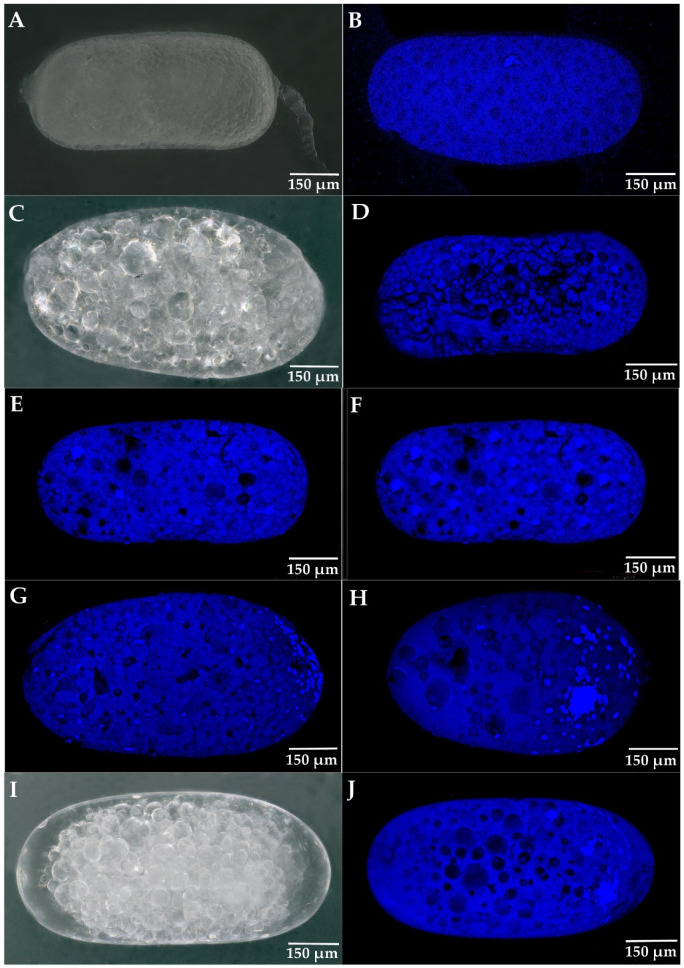
The embryo development process of female–female eggs in *R. flaviceps* (**A**,**C**,**I**): the external form of embryo development (no staining) at various stages of the egg from FF group. (**B**,**D**–**G**,**J**): the external form stained with DAPI at various stages of the egg from FF group. (**A**,**B**): the mature oocyte in ovary. (**B**): mature oocytes with eggshells peeled off; only one egg nucleus (dark blue part). (**C**–**H**): oocyte cleavage stage. (**D**): 0 h after egg laying, there were 3 egg nuclei. (**E**): 5 h after egg laying, there were 7 egg nuclei. (**F**): 48 h after egg laying, there were 34 egg nuclei. (**G**): 5–7 days after egg laying, there were 77 egg nuclei. (**H**): 8–10 days after egg laying, the blastoderm started forming. (**I**,**J**): 10 days after egg laying, the embryo development stopped in the form of the blastocyst, with the yolk cells gathering in the middle of the egg and leaving a hollow space at both ends.

**Figure 7 insects-14-00640-f007:**
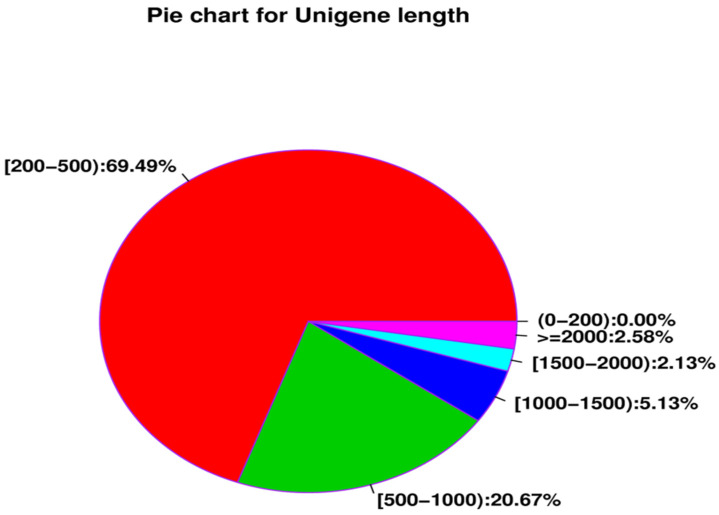
Unigene different length interval pie chart.

**Figure 8 insects-14-00640-f008:**
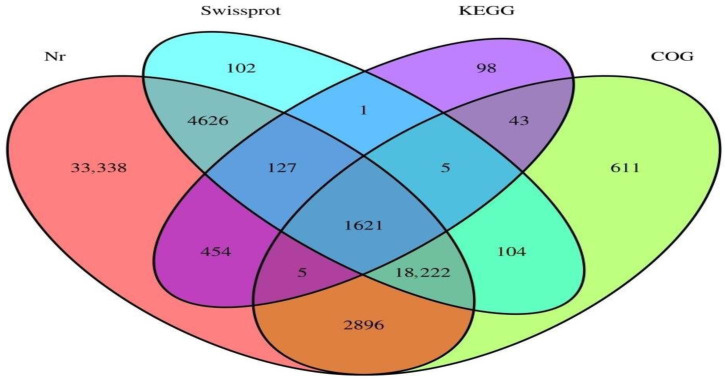
Unigene database annotation results Venn diagram.

**Figure 9 insects-14-00640-f009:**
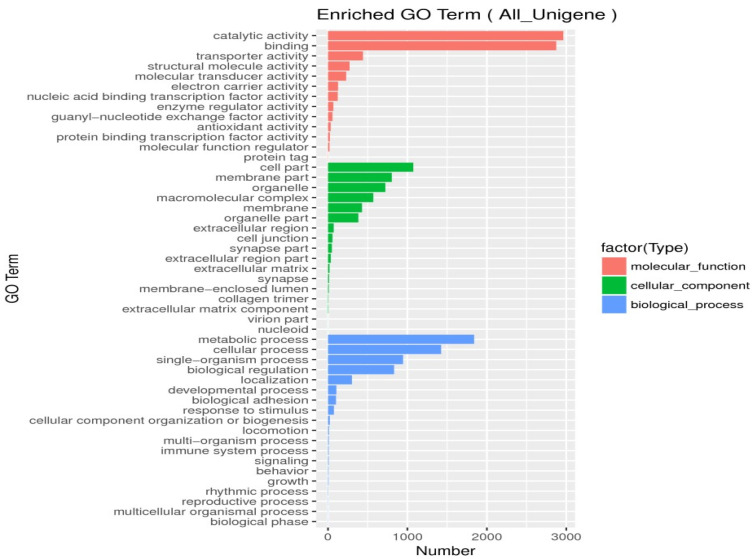
Go function classification. Nucleic-acid-binding guanyl nucleotide exchange protein-binding transcription membrane-enclosed lumen single-organism process multiorganism process.

**Figure 10 insects-14-00640-f010:**
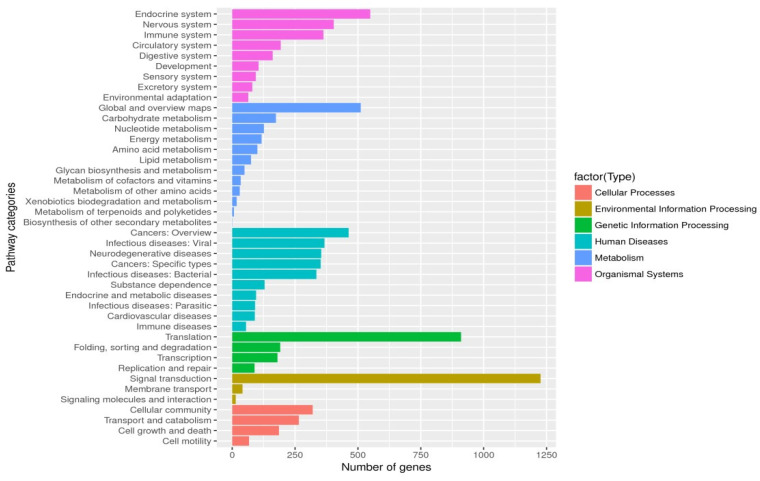
KEGG classification. The ordinate was the secondary classification of the biological pathway, and the abscissa was the number of genes, and the different colors were used to distinguish the primary classification of the biological pathway of parthenogenetic and sexual eggs of *R. aculabialis*.

**Figure 11 insects-14-00640-f011:**
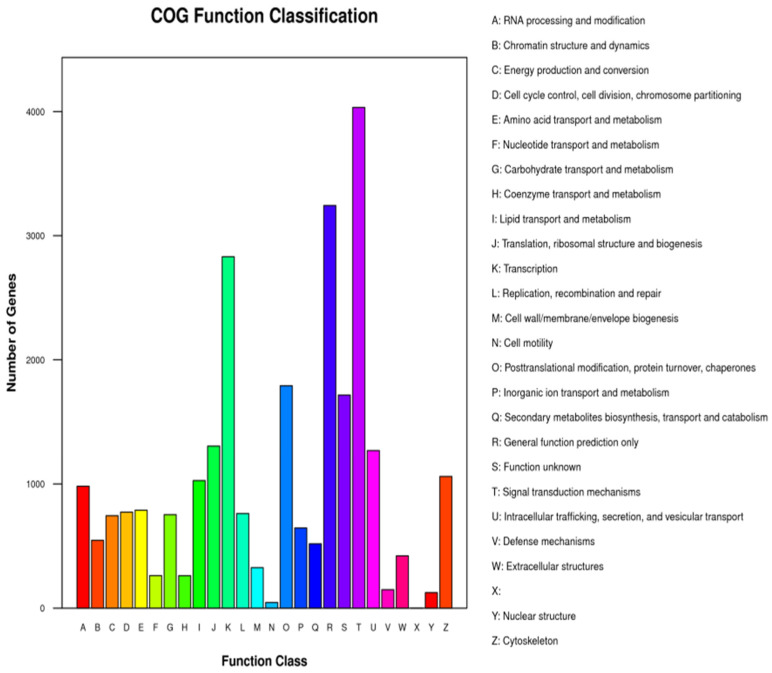
COG function classification of parthenogenetic and sexual eggs of *R. aculabialis*.

**Figure 12 insects-14-00640-f012:**
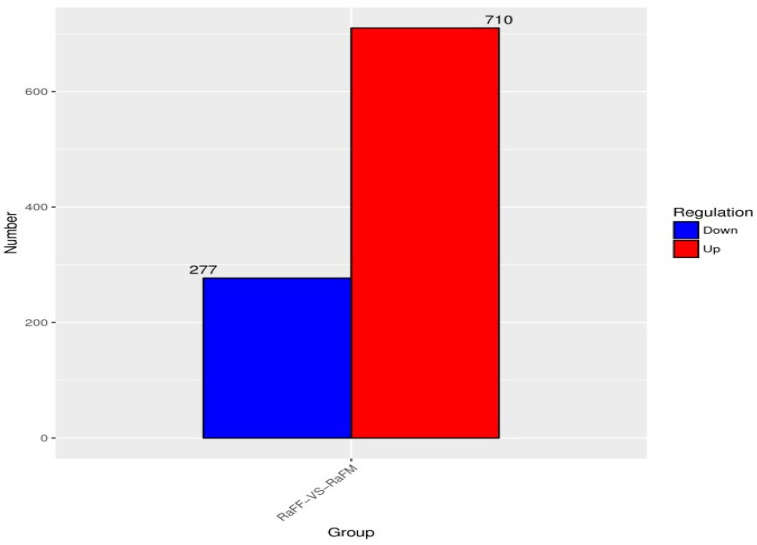
Sample difference comparison of up and down gene expression in RaFF and RaFM.

**Figure 13 insects-14-00640-f013:**
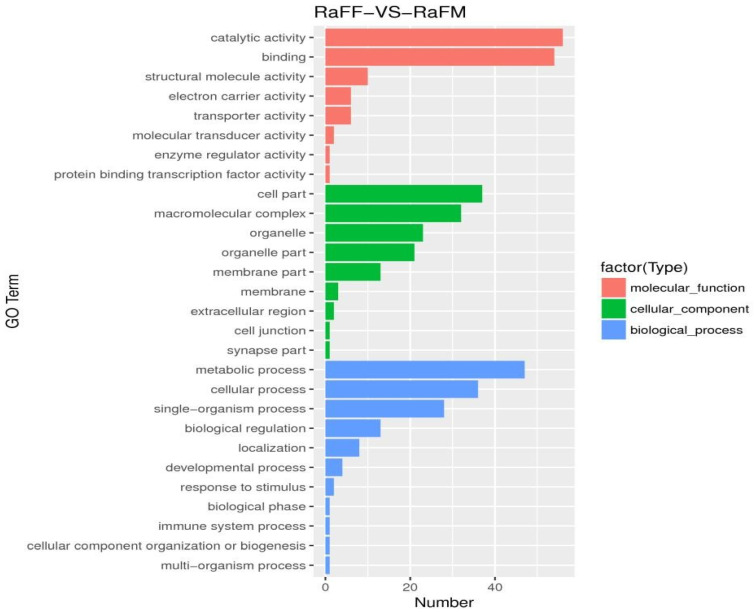
RaFF and RaFM differential gene GO enrichment histogram. The ordinate was the enriched GO term and the abscissa was the number of differential genes in the term. Different colors were used to distinguish between biological processes, cellular components, and molecular functions.

**Figure 14 insects-14-00640-f014:**
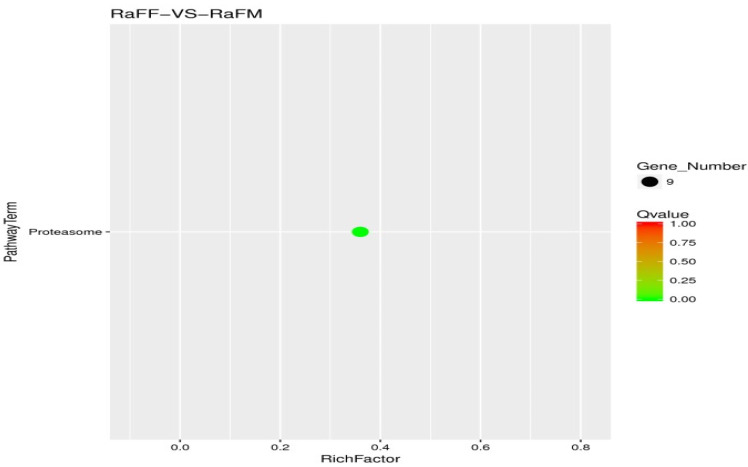
RaFF and RaFM differential gene KEGG enrichment scatter plot.

**Figure 15 insects-14-00640-f015:**
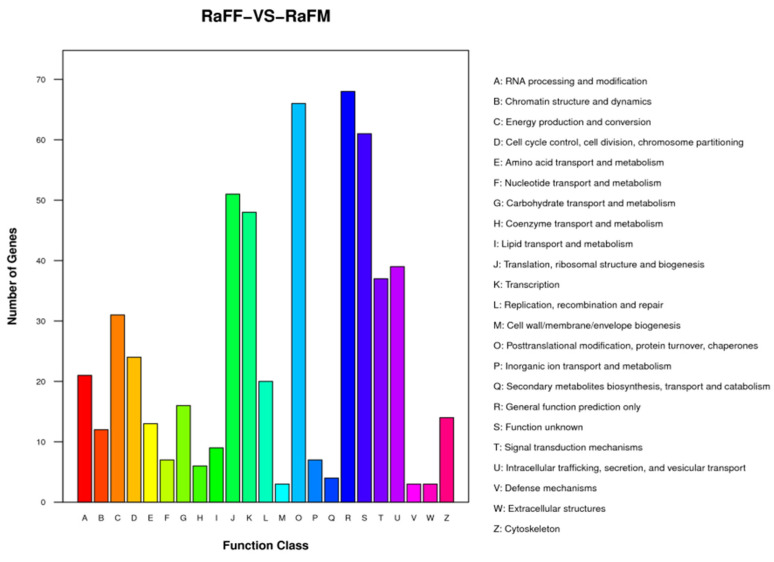
Classification of RaFF and RaFM differential gene COG functions.

**Figure 16 insects-14-00640-f016:**
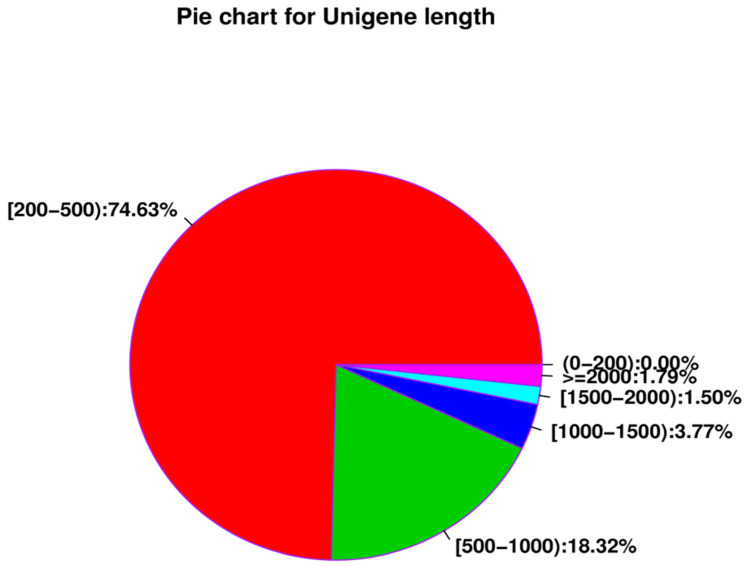
Unigene different length interval distribution pie chart.

**Figure 17 insects-14-00640-f017:**
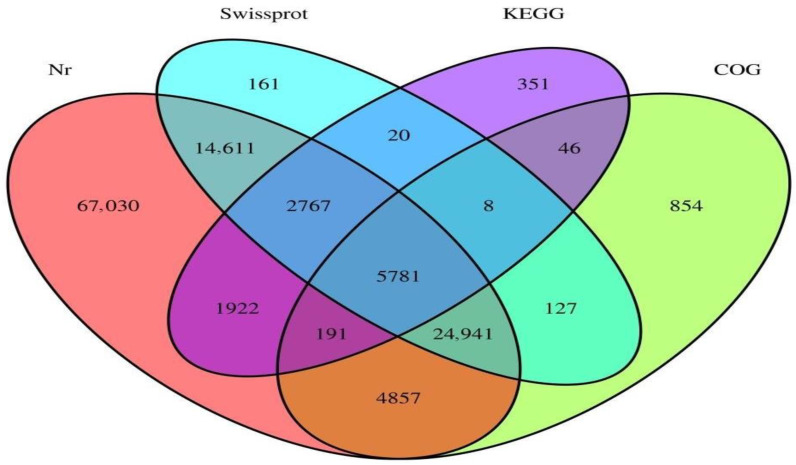
Unigene database annotation results Venn diagram.

**Figure 18 insects-14-00640-f018:**
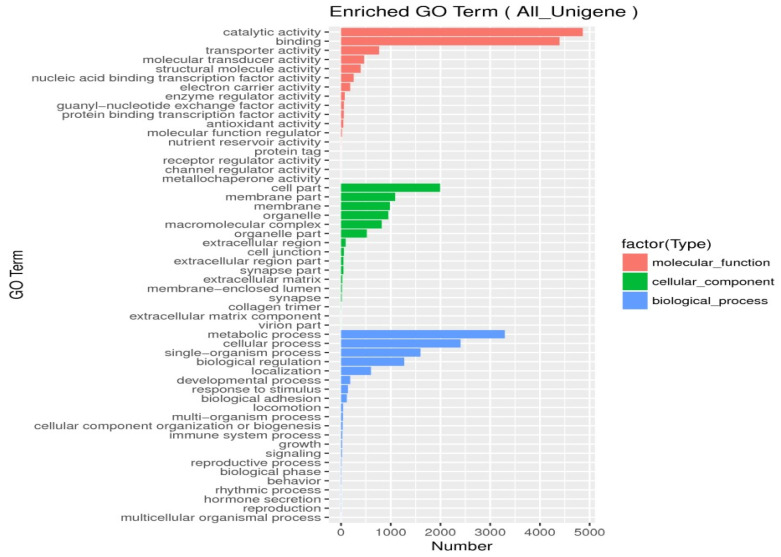
Go function classification. nucleic-acid-binding guanyl nucleotide exchange protein-binding transcription membrane-enclosed lumen single-organism process multiorganism process.

**Figure 19 insects-14-00640-f019:**
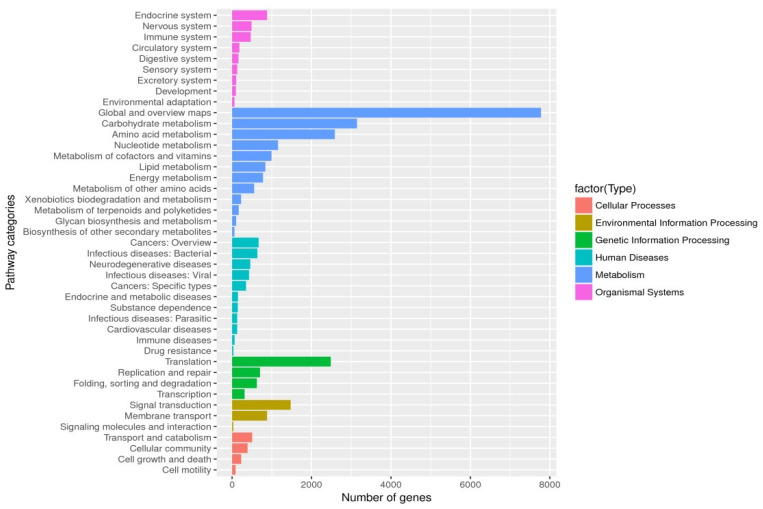
KEGG classification. Folding, sorting, and degradation.

**Figure 20 insects-14-00640-f020:**
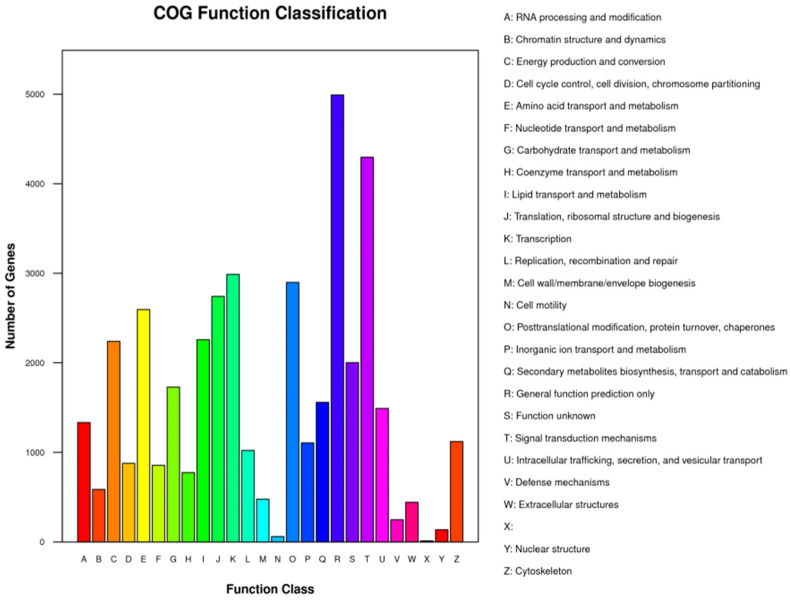
COG function classification.

**Figure 21 insects-14-00640-f021:**
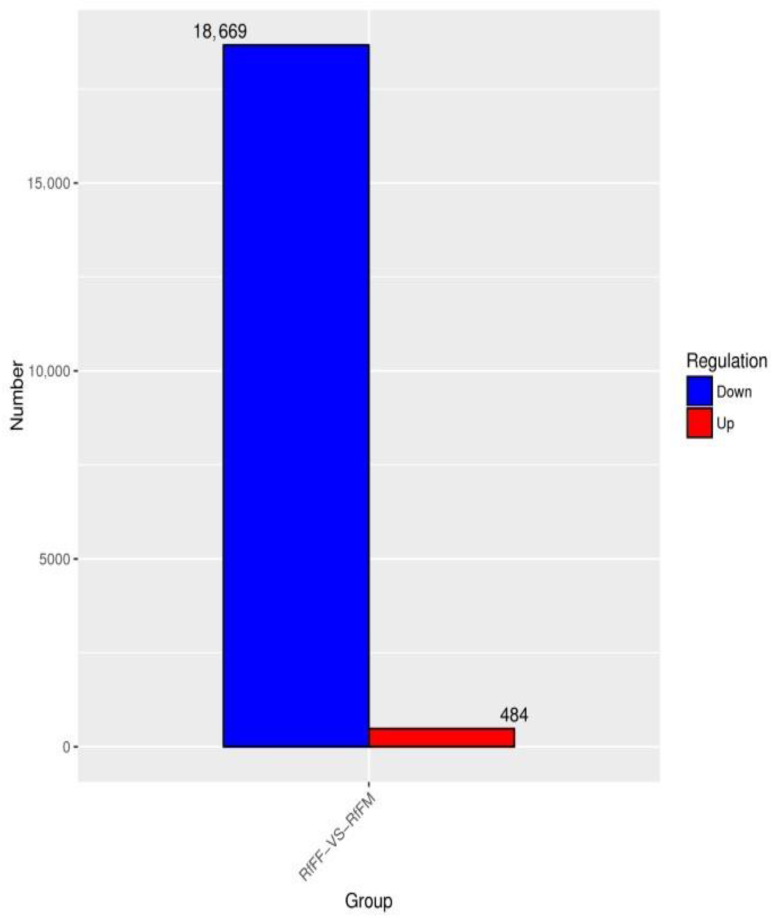
Sample difference comparison of up and down gene expression in RfFF and RfFM.

**Figure 22 insects-14-00640-f022:**
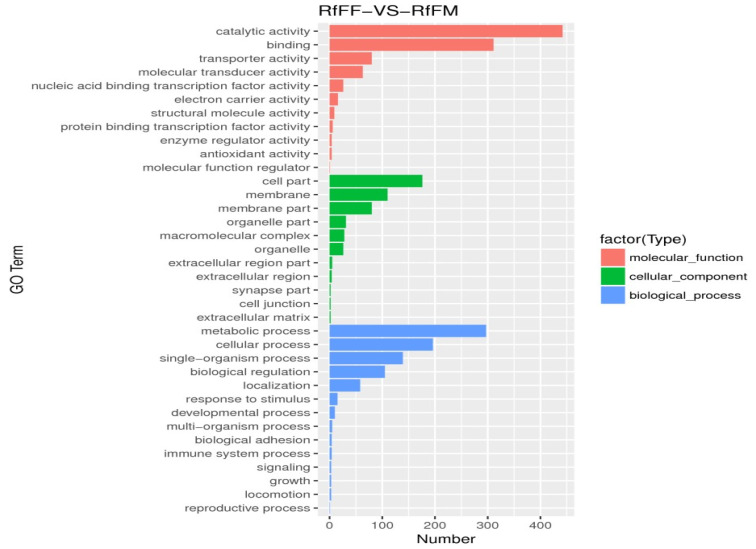
RfFF and RfFM differential gene GO enrichment histogram. The ordinate was the enriched GO term and the abscissa was the number of differential genes in the term. Different colors were used to distinguish biological processes, cellular components, and molecular functions.

**Figure 23 insects-14-00640-f023:**
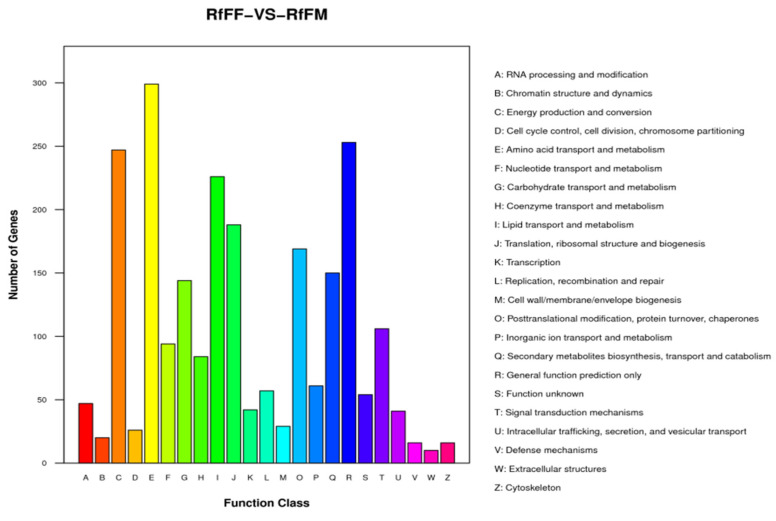
RfFF and RfFM differential gene with the COG functional classification.

**Figure 24 insects-14-00640-f024:**
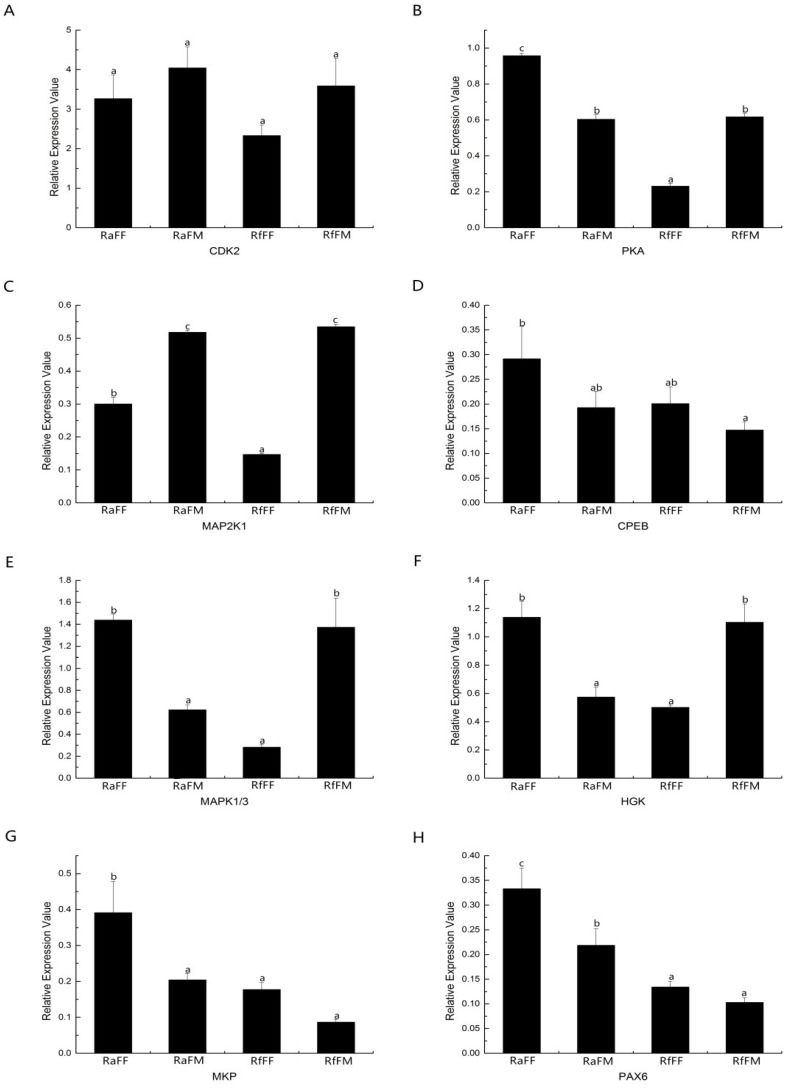
Different expression levels of genes involved in embryonic development. (**A**): the expression levels of the *CDK2* gene in RaFM, RfFM, RaFF, and RfFF. (**B**): *PKA* gene expression levels in RaFF, RfFM, and RaFM was significantly higher than in RfFF. (**C**): *MAP2K1* gene expression levels in RaFM and RfFM were significantly higher than in RaFF. (**D**): expression level of the *CPEB* gene in RaFF was significantly higher than in RfFM. (**E**,**F**): expression levels of *MAPK1/3* gene and HGK gene in RaFF and RfFM were significantly higher than RaFM and RfFF. (**G**): expression level of *MKP* gene in RaFF was significantly higher than in RaFM, RfFM, and RfFF. (**H**): expression level of the *Pax6* gene in RaFF was higher than in RaFM, RfFM, and RfFF. a means significant, b higher significant and most significant (a < 0.05; b < 0.01; c < 0.001).

**Table 1 insects-14-00640-t001:** The gene and primer sequence of RT-PCR.

Short for Gene	Primer Sequences	Annotation
beta-actin	F:CCCAACACAGCGTCTTACAA R:CAGATGTCCTCAGCTTCACG	
*CDK2*	F:ACTCTGTGGTACCGAGCACCTG R:CATGGTTGGCTCCAGCTTCTTCAG	cyclin-dependent kinase 2
*PKA*	F:CTTGGCGCACTTCTCAGTAGACG R:TTGCCTCAACAGACTGGATTGCTG	protein kinase A
*MAP2K1*	F:CCACTAGAACGATGTCGGACCTTC	mitogen-activated protein kinase
	R:CAAGCGCATGGAATTCTTCCTGTG	kinase 1
*CPEB*	F:GCCATCTACCAGGTGTGAAGCAG	cytoplasmic polyadenylation
	R:GGAACGCCAACGGTACGAGTC	element-binding protein
*MAPK1/3*	F:GGCGTATGGAATGGTGGTATCTGC R:TGTTCGCTGGCAGTATGTCTGATG	mitogen-activated protein kinase 1/3
*HGK*	F:TGCAGAGTCAAGGTCTACAGCATG	mitogen-activated protein kinase
	R:TCAAGTCATTCGGTGACCTGATGC	kinase kinase kinase 4
*MKP*	F:CGATGCGGCTGACCTCAACAC	dual-specificity MAP kinase
	R:GCTGCTTGTACGTGATACCACAGG	phosphatase
*Pax6*	F:GTGAAGGCGGAGATGAGATTCTGG R:TCTGTACCTGCACCAAGACCAATG	paired box protein 6

**Table 2 insects-14-00640-t002:** Comparative development of parthenogenetic embryos in *R. aculabialis* and *R. flaviceps*.

Egg Growth, Size, and No	*R. aculabialis*	*R. flaviceps*
Oocyte length at first	605.8 μm	657 μm
Egg phases	Five	Two
Oocyte length at last phase	685 μm	687 μm
Nuclei in mature oocytes	Visible	Few nuclei visible
Before ovulation	Single nucleus	Single nucleus
After oviposition	2–5 nuclei	Not seen
5 h after oviposition	16–22 nuclei	6–10 nuclei
10 h after oviposition	35–42 nuclei	No visible nuclei
24 h after oviposition	51–60 nuclei	No clear nuclei
48 h later	67–84 nuclei	28–37 nuclei
3–5 d after oviposition	Visible blastoderm	
At 5–7 d postoviposition		71–85 nuclei
At 10 d	Blastoderm formation and germinal band growth	86% embryos stopped developing blastoderm formation and 10% stopped developing at the embryonic formation
Embryo rotation	Two time	None
Third stage	Germinal formation	None, large cells concentrated at central point and peripheral sites were free
Fourth and fifth stages	Development of tissues and organs	Lacked any differentiation of appendages and organs development in eggs

**Table 3 insects-14-00640-t003:** Contig and unigene number and length distribution statistics.

Type	Sequences	Bases	Min	Max	Average	N50	(A + T)%	(C + G)%
All_Contig	212,52,163	909,041,501	25	15,261	42.77	39	61.26	38.74
All_Unigene	324,454	170,772,245	201	13,549	526.34	633	60.71	39.29

(1) Type: sequence type. (2) Sequences: total number of unigenes assembled. (3) Bases: total number of bases. (4) Min: shortest unigene length. (5) Max: longest unigene length. (6) Average: the average length of all unigenes. (7) N50: after sorting, unigenes were added in order. When the added length reached half of the total length, the last unigene length was added. (8) (A + T)%: AT base content percentage. (9) (C + G)%: GC base content percentage.

**Table 4 insects-14-00640-t004:** Transcriptome SSR data summary.

Sequence	Quantity
Approved SSR	11,892
Sequence of SSR distribution	11,883
Single nucleotide	9649
Dinucleotide	1236
Trinucleotide	735
Tetranucleotide	258
Pentanucleotide	13
Six nucleotides	1

**Table 5 insects-14-00640-t005:** Contig and unigene number and length distribution statistics.

Type	Sequences	Bases	Min	Max	Average	N50	(A + T)%	(C + G)%
All_Contig	23,146,174	1,012,445,494	25	143,170	43.74	39	58.72	41.28
All_Unigene	422,213	202,603,877	201	42,295	479.86	532	58.13	41.87

(1) Type: sequence type. (2) Sequences: total number of unigenes assembled. (3) Bases: total number of bases. (4) Min: shortest unigene length. (5) Max: longest unigene length. (6) Average: the average length of all unigenes. (7) N50: after sorting, unigenes were added in order. The final unigene length was added when the added length reached half of the total length. (8) (A+ T)%: AT base content percentage. (9) (C + G)%: GC base content percentage.

**Table 6 insects-14-00640-t006:** Transcriptome SSR data summary.

Sequence	Quantity
Approved SSR	7196
Sequence of SSR distribution	7186
Single nucleotide	6033
Dinucleotide	588
Trinucleotide	420
Tetranucleotide	143
Pentanucleotide	10
Six nucleotides	2

## Data Availability

All data generated or analyzed during this study are included in this published article.

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
