# Peer review of "Embryonic Development of Parthenogenetic and Sexual Eggs in Lower Termites"

_insects, 2023, doi:10.3390/insects14070640_

Round 1

Reviewer 1 Report

The introduction, although slightly unorganized, adequately covers all the relevant topics related to this subject. The methodology is comprehensive, providing sufficient details to understand the experimental procedures conducted in this study. The research design also appears appropriate for addressing the study's objectives. The methods are adequately described, offering a clear overview of the experimental procedures and techniques employed.

The cytological results are presented in a clear manner, making them easily understandable. Similarly, the results of the RNAseq analysis are well-presented, providing valuable information on library assembly, unigene length distribution, annotation, functional analysis, and qRT-PCR validation of gene expression.

Moving on to the discussion and conclusion sections, they offer a comprehensive analysis of the research findings, discussing their significance and proposing possible explanations for the observed phenomena. However, I do believe that the discussion section is disproportionately short compared to the result section, which occupies a significant portion of the manuscript. It might be worth considering whether some of the figures and tables could be better suited as supplementary data, allowing for a more balanced and comprehensive discussion section.

I believe this manuscript should be accpeted after some minor corrections.

The manuscript demonstrates a strong understanding of the subject matter; however, there are areas in the text that would benefit from improvement in terms of sentence structure, punctuation, verb tenses, and word choice. Nevertheless, the overall English quality is quite good, and with some minor to moderate English corrections, the clarity and readability of the manuscript can be enhanced.

Reviewer 2 Report

Overall, this is an interesting study with well design and solid data. However, the writhing and analyses of the manuscript require significant modification to be acceptable.

Major points:

1.     What is “lower termites” need to be introduced in the Introduction part and may be the results of this need to be compared with higher termites if have any.

2.     More direct relevant background, reason, hypothesis, rationality and novelty of this study need to be addressed clearly in the Abstract, Introduction or Conclusions. 

3.     This study used a number of techniques, it is good, but the reason and rationality also need to be stated briefly.

4.     The MS is lengthy. Some parts of the methods can be shortened by references citing, particularly sequencing and relevant analyses. Some data, such as Table 1-5, Fig 7-11, 16-20 can be moved to the supplemental files. A more concise words or a table is required for “3.3 The development of parthenogenetic embryos in R. aculabialis”. The Conclusions is too long and some of the words were the repetition of the results. And 3.7. also is too long.

5.     Results, in either development or transcriptome parts, it is better to descript the results of the 2 species together with some comparisons.

6.     In the transcriptome part, a more deep but conclusive and meaningful result is required for FF-vs-FM in 2 species with some clear comparisons, particularly on the gene function. Most words of this part are on general description on the sequencing quality and number of DEGs that from programmatic analysis.

7.     Also in Discussion, a deeper insight based on this study and previous publications is required. Many words are repetitions of Introduction and general information of other species.

8.     The MS needs to be edited by a native English speaker or a professional editing service.

Minor points:

1.       L20, consider to change “simply”.

2.       L55,L62 revise the sentences.

3.       L64, “Hence, studied the embryonic development...”L104, L209... many other places need to be revised. I just provided some samples here.

4.       L172-182, “eggs were cultured for 4 days”, is it still a “single-cell”?

5.       L472, how did you determine they are“parthenogenetic and sexual eggs”

6.       “P”value, please keep consistence by using italic.

See above.

Round 2

Reviewer 2 Report

Yes, Table 1-5, Fig 7-11, 16-20 can be moved to the supplemental files. The new table can be put in the MS.

I still have some doubts about the “parthenogenetic and sexual eggs”. You may clear and can make sure on this, but some more information will be helpful. Such as male-female colonies will all like to mate and produce 100% sexual eggs.

Presentation and language can be improved.

Author Response

Answers to the comments and suggestions of Editors and reviewers

(I)  Revised the entire manuscript according to the referees’ comments.

(III) All the references were checked and they are relevant to the contents of the

manuscript.

(IV) Revisions to the manuscript were highlighted with color changes for easy review by editors and reviewers.

(V)  Sir, every thing is highlighted with a blue color in the the text.

This is a short cover letter that has been provided for the mentioned highlighted  changes in text for the editors’ and referees’ approval. 

Comments on the Quality of English Language

(VI) Yes Sir, our manuscript has been checked by an English colleague who is fluent in English writing as he is a Dutch Entomologist.

OR

If you still suggest us to submit the manuscript for English Editing, then I must submit it for English editing after you finalize this Review Round.

We are agree with you.

(VII) The table 1-5, and Fig 7-11, 16-20 were moved to the supplemental files. The new table was also put in the MS.

(VIII) Sir, I would like to respond on your doubts about “parthenogenetic and sexual eggs”. We set up each colony with two males and one female. Both males provide sperm to females for sexual reproduction. If one of the males fails to transfer sperm for sexual reproduction due to any reason, The other one is healthy and spare enough to do it without any doubt. If the Queen is unfertile, then it fails to lay eggs and produce progeny through sexual reproduction, or parthenogenesis. I hope you will be clear and sure on this, with our defined methodology that was used for the differentiation of sexual reproduction and parthenogenesis.
